# ZoomV: Temporal Zoom-in for Efficient Long Video Understanding

## Abstract

Long video understanding poses a fundamental challenge for large video-language models (LVLMs) due to the overwhelming number of frames and the risk of losing essential context through naive downsampling. Inspired by the way humans watch videos on mobile phones, *constantly zooming in on frames of interest*, we propose **ZoomV**, a query-aware temporal zoom-in framework designed for efficient and accurate long video understanding. Specifically, ZoomV operates in three stages: (1) Temporal interests grounding: guided by the query, ZoomV retrieves relevant events and their associated temporal windows as candidates. (2) Event interests spotlighting: within pools of candidate windows, each window is scored through the model itself reflection and filtered accordingly, where higher-confidence windows are more representative. (3) Compact representation: the selected events are encoded and temporally downsampled to preserve critical semantics while significantly reducing redundancy. Extensive experiments demonstrate that ZoomV substantially outperforms prior **video-agent–style** approaches. On temporal grounding, ZoomV unlocks the latent capability of LVLMs, achieving an **11.8% mIoU** gain on Charades-STA. Remarkably, ZoomV further boosts accuracy on LVBench by **9.7%**, underscoring its effectiveness on long-video benchmarks.

## 1 Introduction

Long-form video understanding, involving frame sequences that can span minutes to hours, presents a fundamental challenge in computer vision. While large video-language models (LVLMs) have shown impressive performance on video-language tasks (Zhang et al., 2024f; Li et al., 2024b; Caba Heilbron et al., 2015; Yu et al., 2019; Lei et al., 2021; Wu et al., 2024), they still struggle with long videos. On one hand, feeding all frames into the model leads to a rapid increase in computational cost and memory usage, making naive full-frame processing infeasible. On the other hand, aggressive temporal downsampling risks discarding critical context, often resulting in severe visual hallucinations. For instance, the advanced LLaVA-Video uniformly samples only 64 frames regardless of video duration, leading to a significant loss of detailed temporal information, especially in hour-long videos (Zhang et al., 2024f). Therefore, it is crucial to identify *a sufficient number of the most relevant frames in a prompt-aware manner* for reliable long-video understanding.

Existing attempts to scale LVLMs to long videos can be mainly grouped into two directions. The first is token sparsification, reducing sequence length by discarding a subset of visual tokens or patches (Li et al., 2023b; Zhang et al., 2024d; Ma et al., 2025; Zhang et al., 2025). While this improves computational efficiency, it inevitably compromises the holistic integrity of frames, and *often leads to noticeable performance degradation*. The second is video-agent approaches, which typically assemble *a pipeline of heterogeneous models*, *e.g.*, (Wang et al., 2024d; 2025b) using EVA-CLIP (Sun et al., 2023) for retrieval, BLIP (Li et al., 2022) for captioning, and GPT-4 (Achiam et al., 2023) for reasoning. Although such modular systems alleviate sequence-length constraints, their reliance on disparate models makes them inefficient, resource-intensive, and non-end-to-end.

To address the limitations of both paradigms, we take inspiration from human cognitive strategies (Sweller, 1994; Zacks & Swallow, 2007; Wu & Xie, 2024; Shen et al., 2024a), particularly the way humans selectively zoom in on relevant visual content, and introduce our method dubbed ZoomV, a query-aware temporal zoom-in framework for efficient long-video understanding. As illustrated in Figure 1 (b), humans review videos broadly to find relevant clues, then gradually focus on more

Figure 1: **Illustration of human-like interaction for long-video understanding.** It divides hour-long videos into manageable sub-events and searches within query-aware segments.

specific sub-events for detailed inspection. Importantly, when the necessary information is not immediately clear, humans may revisit multiple candidate sub-events iteratively, grounding and validating their relevance until they find satisfactory answers. Therefore, ZoomV imitates the behavior and progressively divides the video timeline into coarse-grained events and finer-grained sub-events, enabling efficient human-like search. Specifically, ZoomV unfolds in three progressive stages.

In the first stage, to identify subtle temporal details within promising sub-events accurately, we first need to ground temporal interest. Previous LVLMs (Li et al., 2023a; Zhang et al., 2024f) have incorporated temporal instructions to improve understanding, yet they do not effectively align visual and temporal cues. In contrast, our ZoomV employs a TemporaLink that explicitly embeds temporal information into visual frame representations, enabling LVLMs to precisely associate visual content with corresponding timestamps. Additionally, to alleviate quantization errors introduced by frame sampling, we optimize the absolute timestamp representation, stabilizing temporal learning and enhancing grounding performance. The model retrieves query-relevant events and their associated temporal windows as candidate regions along the video timeline, providing a coarse yet comprehensive coverage of potentially relevant content.

In the second stage, event interests spotlighting, each candidate window is evaluated through the model's self-reflection mechanism dubbed TemporaLight, which assigns confidence scores to spotlight the most representative windows while filtering out less relevant ones. As humans hierarchically search through time, they continuously reflect on whether a specific sub-event warrants deeper inspection. Similarly, we leverage the self-reflection capability of LVLMs to guide the search process. Recent studies (Lin et al., 2022; Kadavath et al., 2022; Zheng et al., 2023; Zhang et al., 2024b) have demonstrated that LLMs effectively assess their prediction confidence through additional multiple-choice or yes/no questions. Inspired by these findings, we first identify that LVLMs inherently possess a similar self-reflection capability—"*they know what they do not know.*" This selective scoring process ensures that only high-confidence sub-events proceed to the next stage.

In the third stage, compact representation, the spotlighted events are encoded and temporally down-sampled into condensed representations that preserve critical semantics while substantially reducing redundancy. Together, these stages enable ZoomV to progressively zoom in from coarse-grained temporal coverage to fine-grained and compact representations, achieving efficient and accurate long-video understanding. Extensive experiments demonstrate its superior performance across various challenging benchmarks, including VideoMME (Fu et al., 2024), MLVU (Zhou et al., 2024), and LongVideoBench (Wu et al., 2024). Notably, on the highly challenging LVBench dataset with hour-long videos (Wang et al., 2024c), ZoomV achieves new state-of-the-art accuracy, substantially surpassing previous methods. ZoomV also substantially outperforms existing video grounding methods on temporal grounding tasks, such as Charades-STA (Gao et al., 2017), ActivityNet Captions (Caba Heilbron et al., 2015), and ReXTime (Chen et al., 2024a). We conduct comprehensive ablation studies and reveal the sources of improvement. In summary, our contributions are threefold:

- We reveal that LVLMs inherently exhibit strong self-reflection abilities, previously studied mainly in LLMs, which enable reflection-guided prioritization of temporal search.

- We propose **ZoomV**, a query-aware hierarchical temporal zoom-in framework that mimics human coarse-to-fine exploration, significantly advancing long-video understanding. For example, ZoomV improves accuracy on LVBench from $41.8\%$ to $51.5\%$.
- We demonstrate that LVLMs possess latent temporal grounding capabilities, which can be effectively unlocked through our proposed TemporaLink. Despite its simplicity, this design achieves an $11.8\%$ mIoU gain over state-of-the-art temporal grounding models.

## 2 RELATED WORK

### 2.1 LONG VIDEO UNDERSTANDING

Understanding lengthy videos for LVLMs is challenging due to the need to store and extract information effectively from numerous frames. One common line involves using language as a bridge to summarize videos into concise captions (Islam et al., 2024; Zhang et al., 2023a), resulting in the omission of vital visual signals. Another widely studied line involves memory-based methods for compressing video features into a limited memory bank, which is achieved by continually updating the memory bank during visual encoding (Song et al., 2024). Memory bank has also been applied to real-time streaming video understanding, potentially enabling an unlimited length of frames while maintaining a constant space footprint (Zhang et al., 2024a). A major drawback of these methods is their oversight of video duration and information density, particularly when utilizing a fixed space for a memory bank. For instance, Flash-VStream compresses both brief 10-second clips and hour-long movies into the same 681 tokens (Zhang et al., 2024a). Besides, these black box methods lack interpretability, as it is hard to verify whether pertinent details are accurately retrieved for reasoning. Another line of work reduces video tokens through visual compression, including spatial token selection (Zhang et al., 2025), temporal frame selection (Yu et al., 2025; Tang et al., 2025), and joint spatio-temporal selection (Shen et al., 2024b). These methods typically serve as pre-processing strategies before LLM inference. In contrast, ZoomV leverages the inherent temporal grounding ability, enabling task-aware and hierarchical selection.

### 2.2 TEMPORAL GROUNDING

Temporal Grounding (TG) localizes video segments relevant to textual queries Gao et al. (2017); Hendricks et al. (2018); Lei et al. (2021) or questions Chen et al. (2024a); Xiao et al. (2024). Current LVLM-based approaches typically rely on heavy customizations to handle continuous time, such as specialized time tokens Wang et al. (2024a); Guo et al. (2024); Ren et al. (2024), auxiliary temporal modules Qian et al. (2024); Wang et al. (2024e), or extensive multi-stage reinforcement training Li et al. (2025a); Wang et al. (2025a). In contrast, ZoomV show that standard LVLMs inherently possess precise grounding capabilities when properly guided. ZoomV unlocks this potential through a simple TemporaLink and coarse-to-fine zooming mechanism, achieving state-of-the-art performance without architectural modifications or sophisticated RL training.

### 2.3 VIDEOAGENTS

VideoAgent is a LLM agent that understands videos by using customized structured and tools (Wang et al., 2024d; Hendricks et al., 2018; Wang et al., 2025b). Previous research usually needs one more model for the agent pipeline. For instance, Wang et al. (2024d) introduces a prompt-driven video QA agent that employs extra vision-language retrieval models (*e.g.*, CLIP) to ground key frames during reasoning. Building on this, VideoTree (Wang et al., 2025b) designs a hierarchical tree-style search by clustering frames with visual features, enabling structured exploration. Both approaches still rely on caption models to provide frame-level descriptions. In contrast, our ZoomV avoids additional captioning or grounding models and directly leverages LVLMs to predict continuous temporal windows. Moreover, the tree-like structure of ZoomV is constructed purely over simple temporal segments, without requiring extra visual feature models or clustering algorithms. We also highlight two concurrent works with hierarchical search strategies. UniTime (Li et al., 2025b) exhaustively performs temporal grounding on all video segments, while VideoChat-R1.5 (Li et al., 2025a) employs a fixed three-step grounding strategy. In contrast, ZoomV designs an *adaptive* search mechanism with reflection-guided early termination: TemporaLight assesses confidence to avoid exhaustive searches and the priority queue enables flexible exploration with backtracking to alternative paths. These

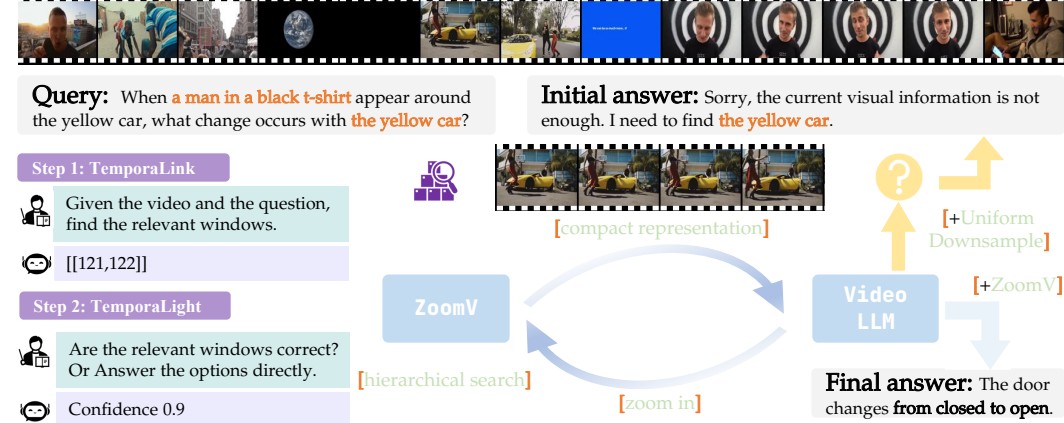

Figure 2: **An illustrative view of ZoomV.** Equipped with ZoomV, Video LLM can gain enhanced capability for efficient and accurate long-video understanding.

innovations allow ZoomV to achieve superior efficiency and accuracy by focusing computation on the most relevant temporal windows.

## 3 PROPOSED APPROACH: ZOOMV

In this section, we present the ZoomV framework, which equips LVLMs with a hierarchical, human-inspired temporal zoom-in mechanism. We first introduce **TemporaLink**, a temporal-augmented representation that explicitly binds timestamps with visual frames to support accurate temporal grounding. Next, we uncover the inherent self-reflection capability of LVLMs and propose **TemporaLight**, and describe the **reflection-guided hierarchical search algorithm**, which integrates grounding and reflection to progressively zoom in from coarse-grained events to fine-grained sub-events. Ultimately, produce compact event representations for efficient long-video understanding.

### 3.1 PRELIMINARY: UNIFIED AUTOREGRESSIVE MODELING

Our ZoomV is built upon an autoregressive LVLM backbone, which sequentially predicts tokens conditioned on visual and textual contexts. An autoregressive LVLM generates an output sequence $\mathbf{y} = (y_1, y_2, \ldots, y_L)$ with length $L$ given a text condition $\mathbf{x}$ and a video condition $\mathbf{v}$ by predicting tokens one at a time based on the previously generated tokens. Assuming that the LVLM is parameterized by $\theta$, the conditional probability distribution of generating a sequence $\mathbf{y}$ given context $\mathbf{x}$ and $\mathbf{v}$ is defined as

$$p_\theta(\mathbf{y}|\mathbf{v}, \mathbf{x}) = \prod_{i=1}^{L} p_\theta(y_i|\mathbf{v}, \mathbf{x}, \mathbf{y}_{<i}), \quad (1)$$

where $\mathbf{y}_{<1} = \emptyset$ and $\mathbf{y}_{<t} = (y_1, y_2, \ldots, y_{t-1})$. Taking video question answering (VQA) as an example, an LVLM predicts the answer distribution $p_\theta(\mathbf{a} \mid \mathbf{v}, \mathbf{q}, I_q)$, where $\mathbf{q}$ denotes the input question and $I_q =$ "Answer the following questions related to this video" serves as the instruction. Here, $\mathbf{v}$ represents a sequence of $T$ downsampled frame tokens extracted from the original video, which are transformed by a dedicated visual encoder and projector into visual tokens. In the following sections, we extend this autoregressive formulation to model both the grounding and reflection mechanisms within a unified framework.

### 3.2 TEMPORAL INTEREST GROUNDING VIA TEMPORALINK

Temporal interest grounding aims to identify the most relevant temporal windows according to the query, modeling continuous numerical timestamps as discrete digit generations (Ren et al., 2024; Guo et al., 2024). The task process is defined as: (1) Given a query $\mathbf{q}$ and the grounding instruction $I_g =$ "Find the relevant windows", the model predicts text sequence $p_\theta(\mathbf{w}|\mathbf{v}, \mathbf{q}, I_g)$; (2) Then the text sequence $\mathbf{w}$ is turned into a set of time ranges $W = [(s_1, e_1), \ldots, (s_K, e_K)]$ with

size $K$, where $s_k, e_k$ signifies the start and end timestamps of $k$-th target window clip. However, LVLMs naturally struggle to accurately handle numerical tasks, especially in temporal tasks involving precise numerical comparisons (Schwartz et al., 2024; Xie, 2024). To alleviate this challenge and effectively activate the inherent temporal grounding capability of LVLMs, we propose a simple yet effective enhancement module, dubbed **TemporaLink**, which explicitly binds timestamps with visual frame representations.

Specifically, given a downsampled video represented by frames $(f_1, f_2, \ldots, f_T)$ and their corresponding fractional timestamps $(t_1, t_2, \ldots, t_T)$, *e.g.*, (0.00, 3.33, 6.67, 10.00), we first round these timestamps to the nearest integer. Then, to ensure a consistent token representation in TemporaLink, we apply left-zero padding, resulting in timestamps like (00, 03, 07, 10):

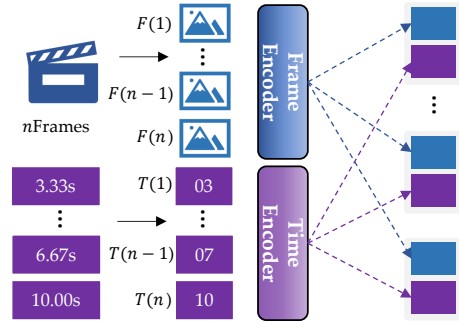

$$\tilde{t}_i = \text{Pad}(\text{Round}(t_i)). \tag{2}$$

Figure 3: **Illustration of TemporaLink.**

Next, we extract frame visual features through a visual encoder $\mathcal{V}$ with a projection module (Zhang et al., 2024f). To embed the absolute timestamp into each frame feature, as illustrated in Figure 3, we directly concatenate these features with their corresponding absolute timestamp embeddings:

$$\tilde{\mathbf{v}}_i = \text{concat}(\mathcal{V}(f_i), \mathcal{T}(\tilde{t}_i)), \; \tilde{\mathbf{v}}_i \in \mathbb{R}^{(N+P) \times D}, \tag{3}$$

where $\mathcal{T}$ denotes the embedding layer of the LLM, $D$ represents the embedding dimension, and $N$, $P$ denote the number of visual frame tokens and padded timestamp tokens, respectively. Following the above design, we perform refinement training on the LVLM. During this process, manually annotated timestamps are further aligned with the rounded timestamps to mitigate quantization errors, as detailed in the Appendix B. By explicitly linking visual frames with timestamps, TemporaLink not only significantly enhances the temporal grounding capability of LVLMs but also provides a solid foundation for the subsequent spotlighting and selection stages.

### 3.3 EVENT INTERESTS SPOTLIGHTING VIA TEMPORALIGHT

After obtaining the candidate temporal windows, we need to validate them and highlight the most suitable ones. Previous research has demonstrated that generative LLMs can evaluate the correctness of their predictions through self-reflection mechanisms (Lin et al., 2022; Zheng et al., 2023; Zhang et al., 2024b;e). These models can produce well-calibrated confidence scores for Yes/No and multiple-choice questions. We extend this observation from text-based LLMs to LVLMs and propose TemporaLight to assess the validity of temporal spotlight predictions. Here, there are two forms of validation.

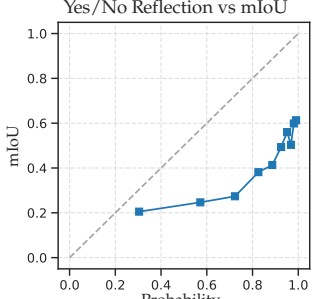

For Yes/No type reflection validation, given a question $\mathbf{q}$, a TemporaLight prediction $W$, and reflection instruction $I_{\text{tf}} =$ "Are the proposed relevant windows correct?", the probability is formulated as

$$c = p_\theta(\text{"Yes"}|\mathbf{v}, \mathbf{q}, W, I_{\text{tf}}). \tag{4}$$

The Yes/No reflection confidence positively correlates with grounding accuracy (mIoU), thus providing an intrinsic measure of spotlight correctness without human annotations (Figure 4, top). For multiple-choice reflection validation, the reflection confidence score is defined by selecting the maximum prediction probability from multiple choices. Given a set of candidate answers, the reflection confidence is computed as:

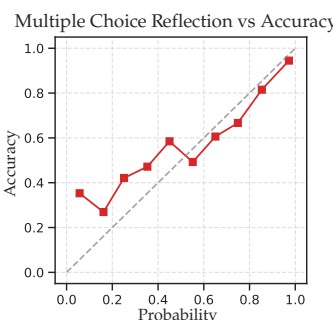

Figure 4: **Reflection probabilities correlate with performance.**

$$c = \max \left\{ p_\theta(o|\mathbf{v}, \mathbf{q}, W, I_{\text{mc}}) \right\}, o \in (\text{"A"}, \text{"B"}, \ldots), \tag{5}$$

**Algorithm 1:** TemporaLight Hierarchical Search

---

**Input:** $\mathbf{v}$, $\mathbf{q}$,
$\Delta$ is the sub-event duration threshold, $\epsilon$ is the confidence threshold.
**1 Initialize:**

- **PQ***: a priority queue prioritised by confidence.*
- *$W$: the candidate optimal window; c: best confidence.*
- $W, c \leftarrow \text{SPOTLIGHTREFLECT}(\mathbf{v})$
- $\text{ENQUEUE}(\mathbf{PQ}, \mathbf{v}, W, priority = c)$

**2 def** $\text{SPOTLIGHTREFLECT}(\mathbf{v}_i)$**:**
**3**     $W_i = \text{GROUND}(\text{FRAMESAMPLE}(\mathbf{v}_i), \mathbf{q}, I_g)$ ;
**4**     **if** *question* $\mathbf{q}$ *is open-ended* **then**
**5**        |   $c_i = \text{REFLECT}(\mathbf{v}, \mathbf{q}, W, I_{\text{tf}})$ // `Yes/No`;
**6**     **else**
**7**        |   $c_i = \text{REFLECT}(\mathbf{v}, \mathbf{q}, W, I_{\text{mc}})$ // `MC`;
**8**     **return** $W_i, c_i$
**9 while PQ** *is not empty* **do**
     // Pop sub-event with top priority;
**10**     $\mathbf{v}_i, W_i, c_i \leftarrow \text{DEQUEUE}(\mathbf{PQ})$ ;
**11**     **if** $c_i \geq c$ **then**
**12**        |   $c \leftarrow c_i$ ;
**13**        |   $W \leftarrow W_i$ ;
**14**     **if** $c_i \geq \epsilon$ **then**
**15**        |   break // stop criterion;
**16**     **for** $\mathbf{v}_j \in \{begin, mid, end\}$ *of* $\mathbf{v}_i$ **do**
**17**        **if** $\text{LENGTH}(\mathbf{v}_j) \geq \Delta$ **then**
**18**           |   $W_j, c_j \leftarrow \text{SPOTLIGHTREFLECT}(\mathbf{v}_j)$;
**19**           |   $\text{ENQUEUE}(\mathbf{PQ}, \mathbf{v}_j, W_j, priority = c_j)$
**Output:** $W$, the optimal temporal windows

---

where $I_{\text{mc}}$ is the reflection instruction, *e.g.*, "`Answer the options directly`". The calibration analysis in Figure 4 (bottom) further confirms that LVLMs produce reliable reflection scores, especially at high-confidence levels. In summary, the above two modes indicate that LVLM inherently knows whether "relevant windows can be found" and "whether questions can be correctly answered."

Notably, ZoomV iteratively performs temporal interest grounding to identify query-aware temporal events. At each step, the TemporaLight module is employed to evaluate the confidence $c$ of the currently candidate windows. If the reflection confidence $c$ is below a predefined threshold $\epsilon$, we hierarchically split the event into three equal-sized overlapping sub-events ("`beginning`", "`middle`", and "`end`".), recursively exploring sub-events prioritized by reflection confidence scores $c$, as illustrated in Algorithm 1. In this algorithm, ZoomV adopts a priority queue **PQ** to organize the order of sub-event searches, which allows backtracking to coarser-grained events to explore alternative search paths when current sub-events still do not yield enough information. This hierarchical search terminates either when the confidence exceeds a threshold hyper-parameter $\epsilon$, or when the sub-event duration falls below a minimal threshold $\Delta$. The highlighted temporal windows with the highest reflection confidence are utilized for the following video understanding tasks.

## 3.4 EVENT COMPACT REPRESENTATION

Receiving the spotlighted events, ZoomV constructs compact representations for them, where high-confidence windows are encoded through the LVLM's visual encoder to extract visual embeddings, followed by temporal downsampling to retain key frames and discard less informative ones. The embeddings are then aggregated into event-level representations that maintain temporal order and salient visual-textual correlations. By concentrating computation on the most representative sub-events, ZoomV achieves effective long-video understanding without excessive memory or latency costs, substantially contributing to its overall performance on challenging benchmarks.

## 4 EXPERIMENTS

### 4.1 EXPERIMENT SETTINGS

**Evaluation Benchmarks.** We comprehensively evaluate our ZoomV across three types of video understanding tasks, including a total of **eight** subtasks. (1) For **video question answering**, ZoomV is validated on three long-video multiple-choice benchmarks, including LongVideoBench (Wu et al., 2024), MLVU (Zhou et al., 2024), and LVBench (Wang et al., 2024c), which cover durations from minutes to hours, as well as short-video benchmarks like MVBench (Li et al., 2024b) and VideoMME (Fu et al., 2024). (2) For **temporal sentence grounding**, we conduct zero-shot evaluation on widely used benchmarks such as Charades-STA (Gao et al., 2017) and ActivityNet-Captions (Caba Heilbron et al., 2015). Following prior works (Ren et al., 2024; Lin et al., 2023), we adopt Recall@1 at IoU thresholds $0.3, 0.5, 0.7$ and mIoU as metrics. (3) For **temporal question grounding** task, we evaluate on the popular ReXTime benchmark (Chen et al., 2024a), which is designed to assess temporal reasoning and causal understanding across multiple video events, and we measure both VQA accuracy and Recall@1 at IoU thresholds $0.3$ and $0.5$.

**Models.** We apply our ZoomV to the LLaVA-Video (Zhang et al., 2024f), InternVL2.5 (Chen et al., 2024b), and advanced Qwen2.5-VL (Team, 2025), three different video model architectures for generality. The training of TemporaLink is both completed within eight hours using 128 A100 GPUs. To enhance TemporaLight capabilities without sacrificing general performance, we apply LoRA (Hu et al., 2022) with a rank of 32 to the LLM, and freeze other parameters.

**Video Input Format.** Within the identified time window $W$, interest frames are densely sampled from the spotlighted segments for video understanding. These dense frames are **appended** after the globally sparsely sampled frames, thereby retaining the ability to answer questions about the global video context. In our experiments, the number of global frames is set to $64$, while the maximum number of spotlight frames provided by ZoomV is 16.

### 4.2 MAIN RESULTS

**Video Question Answering Tasks.** As illustrated in Table 1, ZoomV consistently boosts existing open-source LVLMs across a wide range of benchmarks. On short-video tasks such as MLVU, ZoomV maintains competitive performance, ensuring that its long-video enhancements do not come at the cost of short-duration understanding. The advantage becomes *more pronounced on long-video datasets*: for instance, ZoomV improves InternVL2.5 on LVBench (average duration 4101 seconds) from $41.8\%$ to $51.5\%$, and **makes it surpass all prior methods**. Similar gains are observed on LongVideoBench (+2.7% accuracy) and VideoMME-Long (+1.7%), highlighting its effectiveness in handling videos lasting up to several hours. Furthermore, the consistent improvements across different LVLM backbones, such as an **8.7**% and **11.3**% boost on LVBench with LLaVA-Video and Qwen2.5-VL, demonstrate the robustness and versatility of our ZoomV as a general solution for temporal search in long video understanding.

**Temporal Grounding Tasks.** As shown in Table 2, ZoomV delivers substantial improvements over existing grounding-oriented LVLMs. On standard benchmarks such as Charades-STA and ActivityNet Captions, it achieves an average mIoU gain of **11.8**% compared with the previous state-of-the-art (*e.g.*, GroundedVideo-LLM). Beyond these datasets, ZoomV demonstrates even stronger advantages on ReXTime, where it boosts mIoU by **8.6**%, Recall@0.5 by **9.6**%, and nearly **doubles** VQA accuracy from $40.0\%$ to $76.5\%$ relative to TimeChat, clearly highlighting its ability to reason over complex temporal event structures.

### 4.3 ANALYSIS

**Effectiveness of TemporaLink.** We quantitatively validate TemporaLink by comparing it to the widely used Time Instructions on various datasets. As shown in Figure 5a, compared to the time instruction, TemporaLink significantly improves the ability of moment retrieval, especially at high IoU thresholds. Specifically, when replacing TemporaLink with time instruction on the QVHighlight, R@0.7 drops sharply by $13.8\%$ while R@0.5 drops $6.5\%$. Although previous LVLMs are capable of recognizing relevant events, they struggle to accurately establish associations between

Table 1: **Comparison of ZoomV with other LVLMs on video understanding results.** The results on various short and long video benchmarks with video durations range from seconds to hours.

| Model | Size | #F | MVBench | MLVU | LongVideoBench | VideoMME | | LVBench |
|---|---|---|---|---|---|---|---|---|
| | | | | | | Long | Overall | |
| **Average Duration** | | | 16s | 651s | 473s | 2386s | 1010s | 4101s |
| *Proprietary LVLMs* | | | | | | | | |
| GPT-4V (OpenAI, 2023) | - | - | 43.7 | 49.2 | 60.7 | 53.5 | 59.9 | - |
| GPT-4o (OpenAI, 2024) | - | - | 64.6 | 64.6 | 66.7 | 65.3 | 71.9 | 34.7 |
| Gemini-1.5-Pro (Team et al., 2024) | - | - | 60.5 | 61.8 | 64.4 | 67.4 | 75.0 | 33.1 |
| *Open-Sourced LVLMs* | | | | | | | | |
| InternVL2 (Chen et al., 2024b) | 8B | - | 65.8 | 64.0 | 54.6 | - | - | |
| Qwen2-VL (Wang et al., 2024b) | 7B | - | 67.0 | - | - | - | 63.3 | - |
| Qwen2.5-VL (Bai et al., 2025) | 7B | 768 | - | - | - | - | 65.1 | 45.3 |
| LLaVA-OneVision (Li et al., 2024a) | 7B | - | 56.7 | 64.7 | 56.3 | - | 58.2 | - |
| LLaVA-OneVision (Li et al., 2024a) | 72B | - | 59.4 | 68.0 | 61.3 | - | - | 26.9 |
| *Long-Video LVLMs* | | | | | | | | |
| VideoLLaMA2 (Zhang et al., 2023b) | 7B | 72 | 54.6 | 48.5 | - | 42.1 | 47.9 | - |
| LongVA (Zhang et al., 2024c) | 7B | 128 | - | 56.3 | - | 46.2 | 52.6 | - |
| LLaMA-VID (Li et al., 2023b) | 7B | 1 FPS | 41.9 | 33.2 | - | - | - | 23.9 |
| Oryx (Liu et al., 2024) | 7B | 1 FPS | 63.9 | 67.5 | 55.3 | 50.3 | 58.3 | - |
| Oryx-1.5 (Liu et al., 2024) | 7B | 1 FPS | 67.6 | 67.5 | 56.3 | 51.2 | 58.8 | - |
| LongVU (Shen et al., 2024b) | 7B | 1 FPS | 66.9 | 65.4 | 59.5 | 52.4 | 60.6 | - |
| *Video Agents* | | | | | | | | |
| VideoAgent (GPT-4) (Fan et al., 2024) | - | 87 | - | - | - | 49.0 | 56.0 | - |
| VideoTree (GPT-4o) (Wang et al., 2025b) | - | 98 | - | - | - | 53.1 | - | - |
| UniTime (Li et al., 2025b) | 7B | 128 | - | 66.5 | 56.5 | - | - | - |
| LLaVA-Video (Zhang et al., 2024f) | 7B | 80 | 57.7 | 64.4 | 58.3 | 52.4 | 63.4 | 41.3 |
| *w/* **ZoomV** | | 64+16 | 58.1 (↑ 0.4) | 68.1 (↑ 3.7) | 60.9 (↑ 2.6) | 53.9 (↑ 1.5) | 64.0 (↑ 0.6) | 50.0 (↑ 8.7) |
| InternVL2.5 (Chen et al., 2024b) | 8B | 80 | 70.1 | 67.1 | 60.6 | 52.2 | 63 | 41.8 |
| *w/* **ZoomV** | | 64+16 | 70.3 (↑ 0.2) | 70.0 (↑ 2.9) | 63.3 (↑ 2.7) | 53.9 (↑ 1.7) | 64.4 (↑ 1.4) | 51.5 (↑ 9.7) |
| Qwen2.5-VL (Team, 2025) | 7B | 80 | 66.0 | 65.9 | 59.0 | 52.0 | 63.5 | 40.0 |
| *w/* **ZoomV** | | 64+16 | 66.0 (↑ 0.0) | 67.0 (↑ 1.1) | 61.0 (↑ 2.0) | 53.6 (↑ 1.6) | 63.6 (↑ 0.1) | 51.3 (↑ 11.3) |
| InternVL3 | 8B | 80 | 69.5 | 67.5 | 60.8 | 54.6 | 65.3 | 43.1 |
| *w/* **ZoomV** | | 64+16 | 69.5 (↑ 0.0) | 68.5 (↑ 1.0) | 63.6 (↑ 2.8) | 54.9 (↑ 0.3) | 65.6 (↑ 0.3) | 51.6 (↑ 8.5) |

Table 2: **Comparison of ZoomV with other LVLMs on video grounding results.** The results include two temporal-sentence and one temporal-question grounding benchmarks.

| Model | Charades-STA | | | | ActivityNet-Captions | | | | ReXTime | | | |
|---|---|---|---|---|---|---|---|---|---|---|---|---|
| | R@0.3 | R@0.5 | R@0.7 | mIoU | R@0.3 | R@0.5 | R@0.7 | mIoU | R@0.3 | R@0.5 | mIoU | VQA |
| CG-DETR (Moon et al., 2023) | 70.4 | 58.4 | 36.3 | 50.1 | - | - | - | - | 31.3 | 16.6 | 23.8 | - |
| UniVTG (Lin et al., 2023) | 72.6 | 60.2 | 38.6 | 52.1 | - | - | - | - | 41.3 | 26.8 | 28.1 | - |
| LITA (Huang et al., 2024b) | - | - | - | - | - | - | - | - | 29.49 | 16.29 | 21.49 | 34.44 |
| SeViLA (Yu et al., 2023) | 27.0 | 15.0 | 5.8 | 18.3 | 31.6 | 19.0 | 10.1 | 23.0 | - | - | - | - |
| Valley (Luo et al., 2023) | 28.4 | 1.8 | 0.3 | 21.4 | 30.6 | 13.7 | 8.1 | 21.9 | - | - | - | - |
| VideoChat2 (Li et al., 2024b) | 38.0 | 14.3 | 3.8 | 24.6 | 40.8 | 27.8 | 9.3 | 27.9 | - | - | - | - |
| Momenter (Qian et al., 2024) | 42.6 | 26.6 | 11.6 | 28.5 | 42.9 | 23.0 | 12.4 | 29.3 | - | - | - | - |
| VTimeLLM (Huang et al., 2024a) | 51.0 | 27.5 | 11.4 | 31.2 | 44.0 | 27.8 | 14.3 | 30.4 | 28.8 | 17.4 | 20.1 | 36.1 |
| TimeChat (Ren et al., 2024) | 46.7 | 32.2 | 15.7 | - | - | - | - | - | 14.4 | 7.6 | 11.6 | 40.0 |
| HawkEye (Wang et al., 2024e) | 50.6 | 31.4 | 14.5 | 33.7 | 49.1 | 29.3 | 10.7 | 32.7 | - | - | - | - |
| GroundedVideo-LLM (Wang et al., 2024a) | 54.2 | 36.4 | 19.7 | 36.8 | 46.2 | 30.3 | 19.0 | 36.1 | - | - | - | - |
| Qwen2.5-VL (Bai et al., 2025) | - | 24.2 | 11.1 | 29.0 | - | 15.8 | 7.5 | 21.1 | - | - | - | - |
| VideoChat-R1 (Li et al., 2025a) | - | 70.6 | 47.2 | 59.9 | - | 33.3 | 16.7 | 35.5 | - | - | - | - |
| Time-R1 (Wang et al., 2025a) | 78.1 | 60.8 | 35.3 | - | 58.6 | 39.0 | 21.4 | - | 31.81 | 18.46 | 22.48 | 72.1 |
| UniTime (Li et al., 2025b) | - | 59.1 | 31.9 | 52.2 | - | 22.8 | 14.1 | 27.3 | - | - | - | - |
| **Our ZoomV** | 73.6 | 52.4 | 24.5 | 48.6 | 61.0 | 43.0 | 26.1 | 43.9 | 48.4 | 36.4 | 36.7 | 76.5 |

events and timelines without TemporaLink. Besides, TemporaLink provides consistent improvements on the challenging ReXTime, which requires a strong ability to reason across time.

**Effectiveness of TemporaLight.** To validate our TemporaLight effectiveness, we employ the LLaVA-Video model as the baseline, and equip it with our two types of reflections to watch the difference. The results are reported in Table 3. We find that our TemporaLight enhances the base-

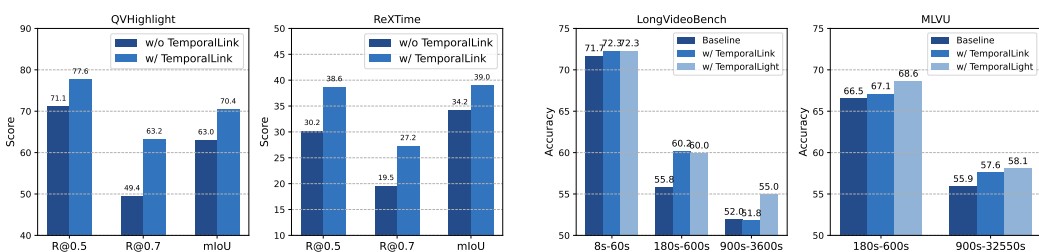

(a) **Effectiveness of TemporaLink.**    (b) **Robustness to video length via TemporaLink.**

Figure 5: **Ablation studies on temporal grounding and ultra-long video length.**

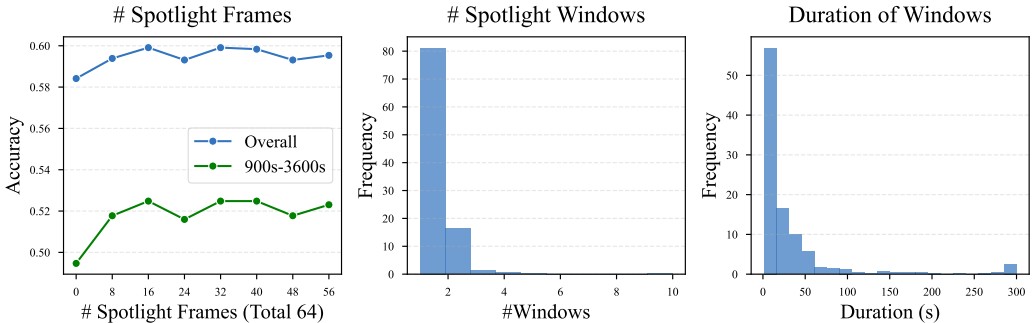

Figure 6: **Impact of the spotlight frames on LongVideoBench.**

line by $8.8\%$ on LVBench, and $2.7\%$ on LongVideoBench. Additionally, multiple-choice reflection, which offers more options for the model, shows better performance on video understanding tasks.

**Ablation Study of Frames Zoomed in.** We conduct experiments by varying the number of frames we zoom in, while keeping the frame budget fixed at $64$. As shown in Figure 6 (left), introducing spotlight frames yields a significant boost in accuracy for both general cases and long videos. Our results suggest that $16$ is an optimal setting, as it preserves global awareness while ensuring precise event retrieval. For a more in-depth understanding, we further analyze the number of spotlight windows and their duration distributions in Figure 6 (middle and right). The histogram of spotlight window counts reveals that most examples require only one or two spotlight windows, suggesting that many questions can be effectively answered with a small number of targeted events. Moreover, the spotlight duration histogram indicates that a majority of spotlighted events are relatively short (under 50 seconds). These findings highlight that a small number of well-chosen short spotlights is sufficient for significant improvements in long-video understanding, validating the effectiveness of our reflection-guided temporal search strategy in selecting relevant video moments efficiently.

**Robustness to Ultra-Long Video Lengths.** ZoomV shows noticeable improvements for video understanding models in Figure 5b. Specifically, for medium-length videos (*i.e.*, `180s-600s`), simply applying the TemporaLink to supplement event details can yield consistent gains. Empirically, despite the significant loss of temporal dynamics in frame sampling, TemporaLink can still identify windows relevant to the questions based on limited visual cues. As the video length increases (*i.e.*, over `900s`), it becomes increasingly challenging for TemporaLink to focus on useful events through sparse frames, and our search strategies are needed and result in significant improvements. Notably, for short videos, the framework maintains original performance as expected.

**Efficiency of ZoomV.** While we boost LVLM performance via ZoomV, we uphold efficiency optimizations to ensure practicality. Initially, training a high-quality ZoomV on the LLaVA-Video model within $80$ epochs only requires $8$ hours utilizing $128$ NVIDIA A100 GPUs. Furthermore, ZoomV introduces minimal additional latency during inference, and the runtime of per search step is $3483$ms. We further optimize the multi-turn search by the prefix cache, as shown in Appendix Table 2. Eventually, we compare our method with other video-agent–style approach (*e.g.*, VideoTree).

Table 3: **Analysis of TemporaLight on video understanding tasks.**

| Method | LVBench | LongVideoBench | LongVideoBench-Long |
|---|---|---|---|
| Baseline (LLaVA-Video) | 41.3 | 58.3 | 48.4 |
| Yes/No Reflection | 43.3 | 61.0 | 50.4 |
| Multiple Choice Reflection | 50.1 | 61.0 | 51.8 |

Table 4: **Detailed Comparison of ZoomV on efficiency and accuracy on EgoSchema.**

| Method | grounding (s) | reflect (s) | caption (s) | keyfr. (s) | QA (s) | overall (s) | acc. (%) |
|---|---|---|---|---|---|---|---|
| VideoTree | – | – | 1.6 | 4.4 | 1.8 | 7.8 | 63.6 |
| ZoomV | 1.6 | 1.9 | – | – | 1.9 | 5.4 | 63.7 |

As shown in Table 4, while achieving higher accuracy, our method requires only 5.4s under the optimal parameters on a typical long-video dataset, compared to 7.8s for VideoTree, yielding an acceleration of approximately 30.8%.

To further validate the efficiency, we compare the inference time of ZoomV with other agents and long-video models across varying durations. As illustrated in Figure 7, we report the runtime of ZoomV in the worst-case scenario (*i.e.*, searching to the finest granularity), benchmarking it against state-of-the-art models including VideoTree (Wang et al., 2025b), LongVU (Shen et al., 2024b), LLaVA-OneVision (Li et al., 2024a), VideoL-LaMA2 (Zhang et al., 2023b), and LLaMA-VID (Li et al., 2023b). ZoomV exhibits a linear scaling of inference time with video duration, yet remains highly efficient. Specifically, for short videos, ZoomV completes inference in just $\sim 5$ seconds, outperforming VideoTree. Even as the duration extends to 1200 seconds, ZoomV requires only $\sim 12.1$ seconds. In sharp contrast, LongVU takes 33 seconds, while VideoLLaMA2 and LLaMA-VID incur significantly higher latency. These results underscore ZoomV's superior scalability and efficiency for long-video understanding.

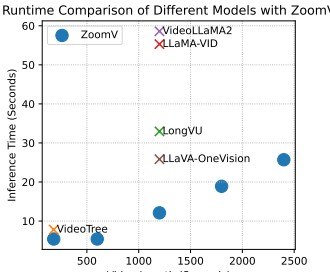

Figure 7: **Runtime comparison across different video durations.** We report the worst-case scenario for ZoomV, demonstrating superior efficiency compared to fixed-cost long-video models.

## 5 CONCLUSION

This paper introduces ZoomV, a novel framework for long-video understanding that emulates a human-like hierarchical temporal search. ZoomV proposes TemporaLink to retrieve key events and TemporaLight to verify predictions and guide the search direction. ZoomV achieves state-of-the-art performance across diverse video benchmarks, demonstrating significant gains in long-video QA and temporal grounding tasks. Furthermore, comprehensive ablation studies confirm the effectiveness of each component and underscore the importance of specialized designs for ultra-long video analysis. Finally, ZoomV bridges the gap between human cognitive strategies and model-based video analysis, providing a robust and interpretable solution for long video tasks.

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

## A   THE USE OF LARGE LANGUAGE MODELS

This paper makes use of an LLM solely for the purpose of polishing paragraph-level text.

## B   ABSOLUTE TIMESTAMP CALIBRATION

As stated in the main text, we utilize quantized integer timestamps to reduce the learning difficulty. However, the frame rate during frame extraction is often low for long videos, while manual annotations are done at a high frame rate. As a result, not every frame corresponding to a manually annotated time can be sampled. For example, the sampled frames are located at `0s,3s,...67s,70s,73s,77s,80s,83s,86s,89s` and the target windows are serialized as `[[72, 82], [84, 89]]`.This problem introduces potential optimization challenges for text-oriented objectives. To address this, we propose the Absolute Temporal Calibration (ATC) method, which precisely aligns the annotated timestamps with the video decoding and frame extraction times. This calibration precisely aligns the annotated timestamps with the video's specific frame time, thereby preventing the model from performing unnecessary frame interpolation during the learning process. Specifically, in the example above, the target windows will first be adjusted to `[[73, 83], [83, 89]]`. Subsequently, we will merge the overlapping windows caused by quantization errors, *i.e.,* calibrated target is `[[73, 89]]`. ATC ensures that the model can focus on temporal understanding without dealing with temporal discrepancies, thereby enhancing the model's learning efficiency and temporal accuracy.

## C   INSTRUCTION TUNING

The objective of Instruction Tuning is to equip the model with the ability to understand the TemporaLink.

Table 1: Various tasks of our instruction dataset with the corresponding number of samples. {r} donate a list of time ranges corresponding to spotlighted video clips.

| Tasks | Sources | Instructions | # of Samples |
|-------|---------|--------------|--------------|
| Spotlight | QVHighlights | Given the video and the query, find the relevant windows. | 7218 |
|  | Grounded-VideoLLM | Provide the timestamps that correspond to the Answer. | 51918 |
| Reflection | ReXTime | Proposed time range: {r}. Is the proposed time range relevant to the question? | 19390 |
|  | Grounded-VideoLLM | Proposed time range: {r}. Is the proposed time range relevant to the question? | 15220 |
| General Answer | Grounded-VideoLLM | General Video-QA instructions | 107806 |
|  | LLaVA-Video | General Video-QA instructions | 79389 |
|  | NextQA | Please respond with only the letter of the correct answer. | 6278 |
| Spotlight Answer | Moment-10M | Please watch the clip of {r} and answer the question. | 42071 |
|  | Grounded-VideoLLM | Please answer the question based on the detailed clip of {r}. | 17214 |

**Datasets**   As shown in Table 1 in the appendix, the training dataset is composed of four distinct tasks, all derived from existing open-source datasets. By introducing specialized instructions, we enhance the model's capabilities in a cost-effective manner. The "Answering" capability is divided into two components: General Answering, which covers basic question-answering tasks like the most of LVLMs, and spotlighted answering, where answers are enriched using grounded video clips identified through a prior search for relevant spotlighted content.

## D   EFFECTIVENESS AND EFFICIENCY TRADE-OFF IN SEARCH

The reflection confidence threshold $\epsilon$ and the minimum sub-event duration $\Delta$ govern the search procedure. These hyperparameters jointly mediate the effectiveness-efficiency trade-off. From an effectiveness perspective, as validated in Figure 1, higher $\epsilon$ and lower $\Delta$ values improve accuracy at the cost of increased search steps. Reducing $\Delta$ from 2400s to 600s with $\epsilon = 0.8$ elevates LVBench accuracy from 46.8% to 49.5%, while finer-grained searches with $\Delta = 300s$ do not result in improvements. Regarding efficiency, the best-case complexity remains constant when $\Delta$ exceeds the

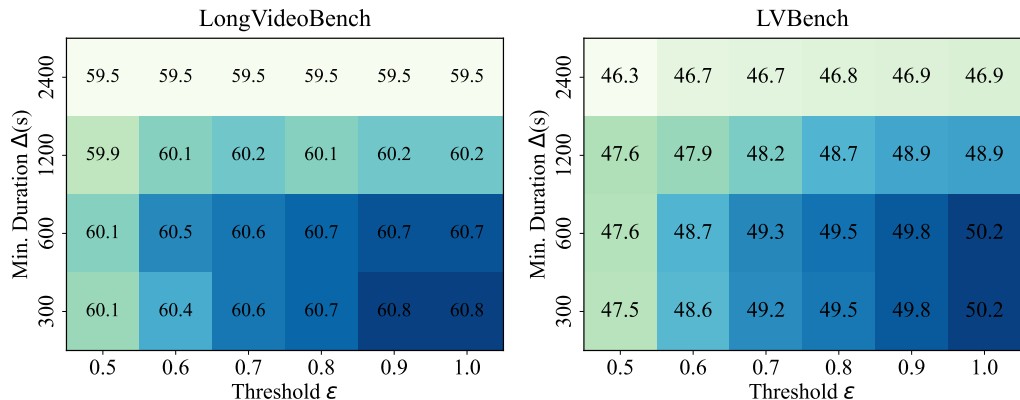

Figure 1: **Effectiveness and efficiency trade-off** with confidence threshold $\epsilon$ and sub-event duration threshold $\Delta$.

Table 2: Optimization of ZoomV in Efficiency via prefix cache.

| Primitives | encoded frames | prefill (ms) | decode (ms) | overall (ms) |
|---|---|---|---|---|
| Grounding | 64 | 1157 | 424 | 1581 |
| Reflection | 80 | 1496 | 406 | 1902 |
| Reflection (w/ prefix cache) | 16 | 299.2 | 406 | 745 |

video length, while the worst-case complexity scales linearly. Specifically, when $\Delta$ is larger than the video length, the search only executes a single step. In contrast, when $\Delta$ is smaller than the video length, setting $\epsilon = 1$ forces exhaustive traversal of all sub-events. The search prioritizes high-confidence segments through a priority queue, emulating human-like coarse-to-fine understanding. Empirical experiments demonstrate that $\epsilon = 0.5$ requires only an average of 1.6 search steps while maintaining 99.5% of peak accuracy on LongVideoBench when $\Delta = 1200s$.

# E    QUALITATIVE ANALYSIS

To further illustrate how **ZoomV** addresses challenges inherent in long-video understanding, we conduct a series of case studies on tasks involving temporal perception and chronological relations Wu et al. (2024). A core difficulty for LVLMs lies in insufficient temporal details, which often leads to misinterpretations of events. ZoomV mitigates this issue by integrating human-like *Spotlight* and *Reflection* mechanisms, allowing for more precise event retrieval.

For example, Figure 2 illustrates a case spanning 275 seconds, in which a man is sitting in front of a mirror. At the global (coarse) sampling level, only sparse frames can be observed, making it difficult to discern the subtle motion of his hands. The TemporaLink component in ZoomV addresses this issue by spotlighting a more fine-grained window from the 249th to the 275th second. Within this localized segment, the frame rate is increased, revealing that the man's hands are clasped together—an action easily missed under low-frequency sampling. This example demonstrates how TemporaLink adaptively zooms in on the essential moments of a long video, capturing subtle actions that would otherwise be overlooked. Additionally, Figure 3 and Figure 4 showcase object attribute change and appearance order cases.

Figure 5 illustrates how ZoomV discerns sequential relationships between events in an ultra-long video through a hierarchical, coarse-to-fine search. In this example, ZoomV first identifies a large time window that roughly contains the relevant events. Upon noticing the disappearance of the white car, the search narrows to the 600-second sub-event window. Within this finer scope, ZoomV uses spotlight frames to focus on critical moments, identifying the appearance of a red car and a person. By progressively refining, ZoomV effectively captures the sequential flow of events, mimicking the way humans would search through long videos by zooming in on key events.

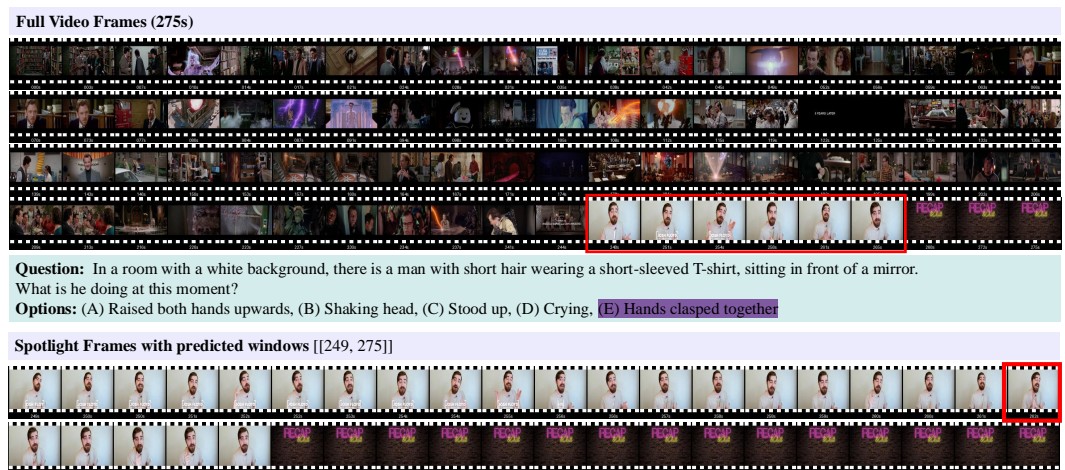

Figure 2: **Illustration of the *subtle temporal dynamic* challenge.**

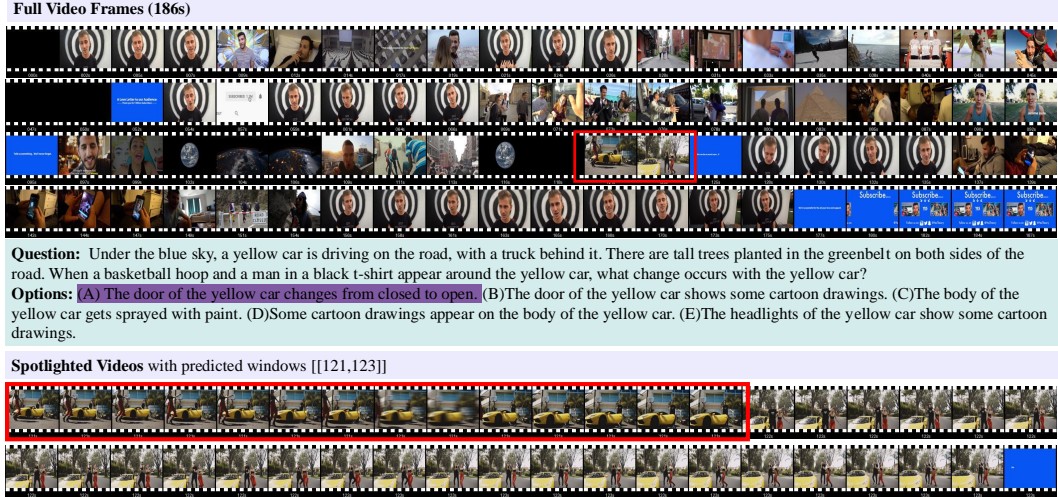

Figure 3: **Illustration of the *object attribute change* challenge.**

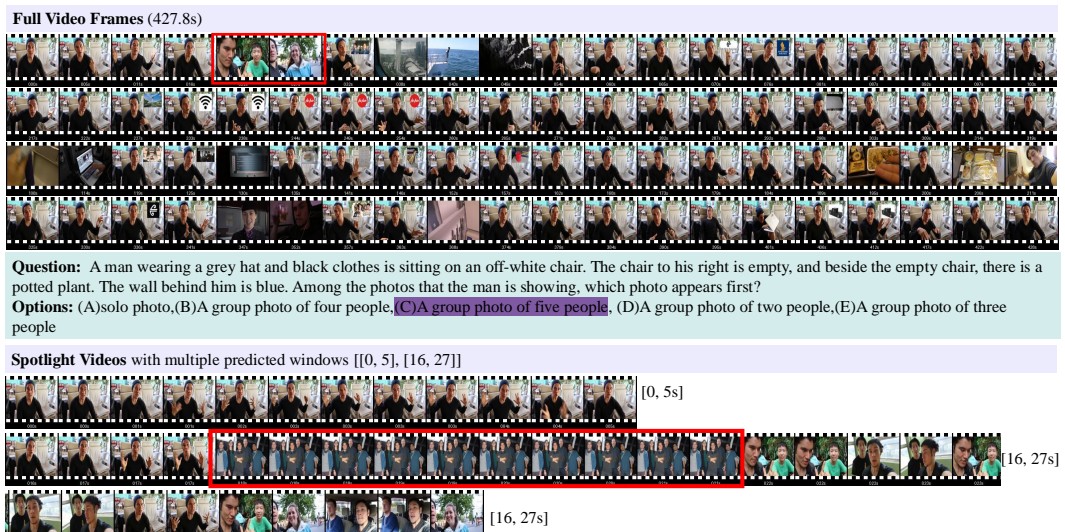

**Full Video Frames (427.8s)**

**Question:** A man wearing a grey hat and black clothes is sitting on an off-white chair. The chair to his right is empty, and beside the empty chair, there is a potted plant. The wall behind him is blue. Among the photos that the man is showing, which photo appears first?
**Options:** (A)solo photo,(B)A group photo of four people,(C)A group photo of five people, (D)A group photo of two people,(E)A group photo of three people

**Spotlight Videos** with multiple predicted windows [[0, 5], [16, 27]]

Figure 4: **Illustration of the *object before/after object* challenge.**

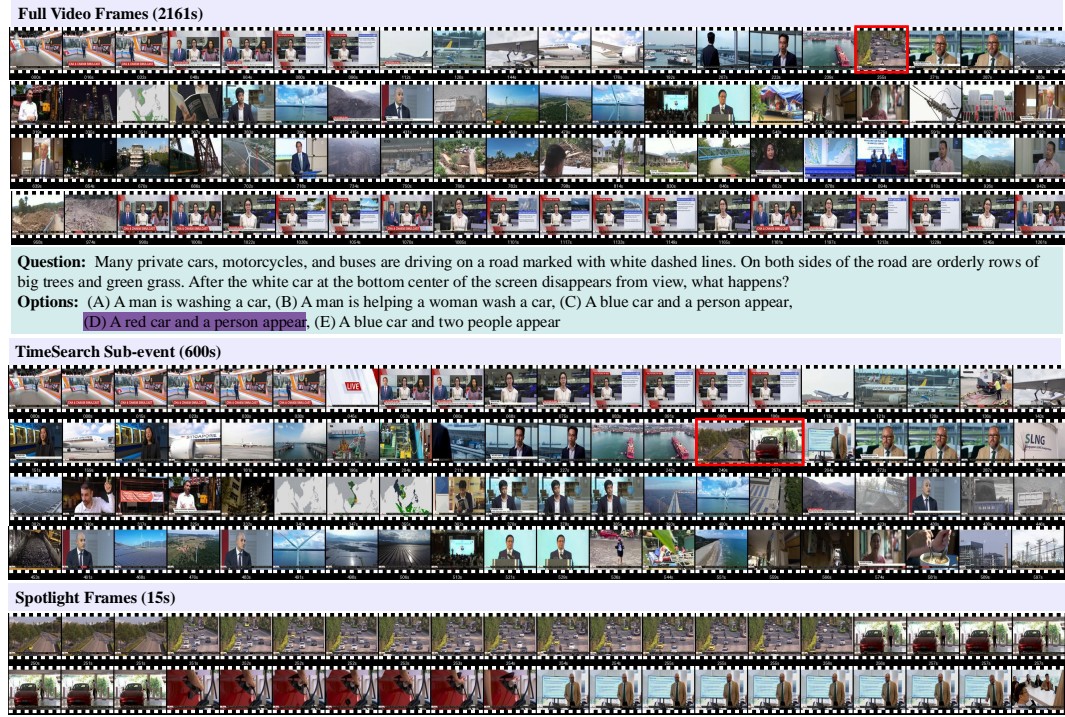

**Full Video Frames (2161s)**

**Question:** Many private cars, motorcycles, and buses are driving on a road marked with white dashed lines. On both sides of the road are orderly rows of big trees and green grass. After the white car at the bottom center of the screen disappears from view, what happens?
**Options:** (A) A man is washing a car, (B) A man is helping a woman wash a car, (C) A blue car and a person appear, (D) A red car and a person appear, (E) A blue car and two people appear

**TimeSearch Sub-event (600s)**

**Spotlight Frames (15s)**

Figure 5: **Illustration of the *event after event* challenge.**

