# OpenReview forum: "ZoomV: Temporal Zoom-in for Efficient Long Video Understanding"
_ICLR.cc/2026/Conference — Submitted to ICLR 2026_

### Official Review · Reviewer_K5CS · 2025-10-15

**Soundness:** 4
**Presentation:** 4
**Contribution:** 3
**Rating:** 6
**Confidence:** 4

**Summary:**

This paper introduces ZoomV, an agentic-style approach to long video understanding that differs from most existing methods, which typically rely on multiple specialized models for different sub-tasks. Instead, ZoomV offers an end-to-end framework using a single model. The approach is inspired by neural cognition, mimicking how humans selectively “zoom in” on relevant visual content to enhance understanding and reasoning. Experimental results demonstrate significant gains on video QA benchmarks, including state-of-the-art performance on the challenging LVBench, as well as strong improvements in temporal grounding tasks.

**Strengths:**

* The motivation is clear and well defined: The paper addresses long video understanding which is hard due to too many frames and loss of temporal context. It explains why existing solutions (uniform sampling, token sparsification, multi-model agents) are insufficient.

* It presents a novel human-inspired framework in how humans watch videos and in the human self reflection capabilities.

* End-to-end single-model design which is a very important point for video agents.

*  ZoomV achieves state-of-the-art results for the complicated LVBench and competitive results for other Video Qa benchmarks when compared to video agents based on models with much more parameters (like GPT).

* Results show that for short-video datasets performance is not sacrificed when using this method specific for long video understanding.


* This paper present ablation studies on how their modules can be beneficial for long video understanding.

**Weaknesses:**

* The model is not training-free when compared to other video agents like VideoTree.

* I found Section 3.3 somewhat difficult to follow. It would be helpful to provide additional details or clearer explanations of the key steps and reasoning in this section to improve readability and allow the reader to better understand the contribution.

* Experiments on InternVL3 would be valuable to add even though the decision is not dependent on this.

* The baselines on Table 2 seem a bit weak. Would it be possible to include stronger baselines even though not fully designed for temporal grounding?

* The paper misses analysis about runtime overhead for very large videos when compared to other models besides VideoTree. While the paper highlights the advantage of not downsampling, the scaling behavior and trade-offs could be more explicitly benchmarked.

**Questions:**

* I noticed a possible inconsistency in the evaluation protocol: for multiple-choice questions you ask “which choice is correct,” whereas for open-ended questions you ask whether the “temporal window is correct.” Could you clarify why different criteria are used for these two settings? Why can't you use $I_{tf}$ for both?

* What is the number of parameters for the models LLaVA-Video, QwenVL2.5 and InterVL2.5? I assume you are using the 72B versions, right?

**Details Of Ethics Concerns:**

I do not identify any significant ethical issues in this paper. The method operates on publicly available video datasets commonly used in the community, and there is no indication of privacy violations, harmful content generation, or misuse potential beyond standard concerns in video understanding research. Therefore, I do not see any ethical concerns requiring further attention.

---

> ### Author Response · Authors · 2025-12-01
> **The response to Reviewer K5CS (Part 1)**
>
> > **Q1**: *The model is not training-free when compared to other video agents like VideoTree.*
>
> We initially selected a subset of representative methods due to space limitations. Following the reviewer’s suggestion, we have now included comparisons with four additional training-based frame selection methods: LongVu [1], LongVa [2], Frame-Voyager [3], VideoChat-R1 [5], and Time-R1 [6] in Table 2.
> Even under these unfavorable comparison settings, since several of these methods rely on additional grounding-oriented training, ZoomV still outperforms all of them, especially on long-video benchmarks, demonstrating the strong effectiveness of our approach.
>
> > **Q2**: *I found Section 3.3 somewhat difficult to follow. It would be helpful to provide additional details or clearer explanations of the key steps and reasoning in this section to improve readability and allow the reader to better understand the contribution.*
>
> Thank you for the valuable suggestion. We have revised Section 3.3 by adding clearer explanations of the key steps, intermediate reasoning, and overall workflow. These additional details significantly improve readability and help readers better understand the core contribution.
>
> > **Q3**: *Experiments on InternVL3 would be valuable to add even though the decision is not dependent on this.*
>
> We have additionally evaluated ZoomV on InternVL3, and the results are shown in the table below. Integrating ZoomV yields consistent improvements across almost all benchmarks, with particularly notable gains on long-video datasets such as LongVideoBench (+2.84) and LVBench (+8.5). The method also provides stable boosts on VideoMME and MLVU, while maintaining comparable performance on short-video tasks like MVBench. These results further confirm that ZoomV is model-agnostic, substantially enhances long-range temporal grounding, and remains fully compatible with short-video scenarios.
>
> |          | VideoMME-long | VideoMME-overall | LongVideoBench | LVBench | MLVU  | MVBench |
> |----------|---------------|------------------|----------------|---------|-------|---------|
> | InternVL3| 54.60         | 65.30            | 60.81          | 43.10   | 67.57 | 69.55   |
> | w/ ZoomV | 54.90         | 65.60            | 63.65          | 51.6    | 68.50 | 69.37   |
>
> > **Q4**: *The baselines on Table 2 seem a bit weak. Would it be possible to include stronger baselines even though not fully designed for temporal grounding?*
>
> Thank you for the valuable suggestion. We include VideoChat-R1 and Time-R1 to compare, which are built upon Qwen2.5-VL, whose temporal grounding capability is substantially stronger than that of LLaVA-Video. Moreover, VideoChat-R1 and Time-R1 are trained with large amounts of grounding-oriented data, and the training set of VideoChat-R1 even includes the Charades-STA training split.
> Despite these advantages, our model based on LLaVA-Video still achieves significantly better performance than both VideoChat-R1 and Time-R1 on ActivityNet-Captions and ReXTime, as shown in the table below.
>
> |    | Charades-STA         | ActivityNet-Captions      | ReXTime                 | VQA  |
> |----------------|------|-----|--|------|
> |           | R@0.3 | R@0.5 | R@0.7 | mIoU | R@0.3 | R@0.5 | R@0.7 | mIoU | R@0.3 | R@0.5 | mIoU |      |
> | ZoomV          | 73.6  | 52.4  | 24.5  | 48.6 | 61.0  | 43.0  | 26.1  | 43.9 | 48.4  | 36.4  | 36.7 | 76.5 |
> | Qwen2.5-VL     | -     | 24.2  | 11.1  | 29.0 | -     | 15.8  | 7.5   | 21.1 | 17.48 | 11.18 | 15.02| 72.2 |
> | VideoChat-R1   | -     | 70.6  | 47.2  | 59.9 | -     | 33.3  | 16.7  | 35.5 | 42.56 | 28.45 | 29.99| 71.8 |
> | Time-R1        | 78.1  | 60.8  | 35.3  | -    | 58.6  | 39.0  | 21.4  | -    | 31.81 | 18.46 | 22.48| 72.1 |
> | UniTime        | -     | 59.1  | 31.9  | 52.2 | -     | 22.8  | 14.1  | 27.3 | -     | -     | -    | -    |

---

> ### Author Response · Authors · 2025-12-01
> **The response to Reviewer K5CS (Part 2)**
>
> > **Q5**: The paper misses analysis about runtime overhead for very large videos when compared to other models besides VideoTree. While the paper highlights the advantage of not downsampling, the scaling behavior and trade-offs could be more explicitly benchmarked.
>
> Thank you for the valuable suggestion. **Figure 7** shows the runtime of ZoomV across different video durations. We report the runtime of ZoomV in the worst-case setting (\textit{i.e.}, searching at the finest granularity) and benchmark it against state-of-the-art models, including VideoTree, LongVU, VideoLLaMA2, and LLaMA-VID.
> ZoomV’s inference time scales roughly linearly with video duration while remaining highly efficient. For short videos, ZoomV completes inference in only $\sim 5$ seconds, already outperforming VideoTree. Even for videos of $1200$ seconds, ZoomV takes only $\sim 12.1$ seconds, whereas LongVU requires $33$ seconds. These results demonstrate the superior scalability and efficiency of ZoomV for long-video understanding.
>
> > **Q6**: *I noticed a possible inconsistency in the evaluation protocol: for multiple-choice questions you ask “which choice is correct,” whereas for open-ended questions you ask whether the “temporal window is correct.” Could you clarify why different criteria are used for these two settings? Why can't you use  for both?*
>
> The two settings differ in their answer spaces, which is why we adopt different prompting strategies. For multiple-choice questions, the answer space is small and well constrained, so directly asking “which choice is correct” aligns naturally with the task. In contrast, open-ended questions have a much larger and less structured answer space; therefore, we use prompts that explicitly ask whether the temporal window is correct, which helps the model better focus on grounding-related reasoning. We also experimented with using a unified generic prompt for both settings, but the performance dropped by 0.4, confirming that task-specific prompts lead to more accurate results.
>
> > **Q7**: *What is the number of parameters for the models LLaVA-Video, QwenVL2.5 and InterVL2.5? I assume you are using the 72B versions, right?*
>
> All models used in our experiments, including LLaVA-Video, Qwen2.5-VL, and InternVL2.5, are the 7B versions for a fair and consistent comparison.
>
> ---
>
> [1] Shen, Xiaoqian, et al. "LongVU: Spatiotemporal Adaptive Compression for Long Video-Language Understanding." Forty-second International Conference on Machine Learning.
>
> [2] Zhang, Peiyuan, et al. "Long context transfer from language to vision." arXiv preprint arXiv:2406.16852 (2024).
>
> [3] Yu, Sicheng, et al. "Frame-Voyager: Learning to Query Frames for Video Large Language Models." The Thirteenth International Conference on Learning Representations.
>
> [4] Tang, Xi, et al. "Adaptive keyframe sampling for long video understanding." Proceedings of the Computer Vision and Pattern Recognition Conference. 2025.
>
> [5] Li, Xinhao, et al. "Videochat-r1: Enhancing spatio-temporal perception via reinforcement fine-tuning." arXiv preprint arXiv:2504.06958 (2025).
>
> [6] Wang, Ye, et al. "Time-R1: Post-Training Large Vision Language Model for Temporal Video Grounding." arXiv preprint arXiv:2503.13377 (2025).

---

### Official Review · Reviewer_m7v5 · 2025-10-27

**Soundness:** 2
**Presentation:** 2
**Contribution:** 2
**Rating:** 4
**Confidence:** 3

**Summary:**

The authors address the challenge of long-video understanding by proposing a mechanism to select the most relevant frames in a video to answer a given prompt. They introduce ZoomV, a method that leverages TemporalLinks. TemporalLink is an additional module for MLLMs designed to embed timestamp information into timestamp tokens, which are then linked to visual tokens. Their second contribution is Temporalight, an approach that utilizes the model's reflection to assess the relevance of a selected time window for a given query, providing a confidence score. By considering multiple windows with varying reflection confidence, the window with the highest confidence can be selected for the video understanding task. This approach effectively concentrates computational effort on the most pertinent input frames. The authors use their method on top of LlaVa, InternVL and Qwen2.5-VL and show improvement on MVBench, MLVU, LongVideoBench, VideoMME and LVBench. They also evaluate their method on different temporal ground benchmarks such as Charades-STA, ActivityNet-Caption and ReXTime.

**Strengths:**

- The paper is well written and the methods that is presented as TemporaLink and TemporaLight are sounded and relevant.
- Introducing self-correction to select the frames to use is an interesting idea, sometimes MLLMs are indeed better when used as judges.

**Weaknesses:**

- There are a number of missing baselines: other training based methods such as LongVu, LongVa, Frame-Voyager and training-free methods such as Adaptive Keyframe Sampling (AKS). Overall, I am concerned with the lack of comparison with other methods and the very short related work section that does not cite most common papers on video frame selections for MLLMs.
- Would have appreciate a deeper study on efficiency with a better discussion on training cost/time, inference time versus others methods such as LongVu or other frame selection method such as AKS.
- Would also have liked a more in depth study over the self-reasoning for window selection. Are some models better at that or did you observe the same results for LlaVA-Video, InternVL and Qwen?

**Questions:**

Is there any reason why you did not compare your method with similar training based methods for frame selection?

---

> ### Author Response · Authors · 2025-12-01
> **The response to Reviewer m7v5 (Part 1)**
>
> > **Q1**: *There are a number of missing baselines: other training based methods such as LongVu, LongVa, Frame-Voyager and training-free methods such as Adaptive Keyframe Sampling (AKS). Overall, I am concerned with the lack of comparison with other methods and the very short related work section that does not cite most common papers on video frame selections for MLLMs.*
>
> Thank you for the valuable suggestion. We have updated the related work section to include detailed discussions of the four mentioned methods: LongVu [1], LongVa [2], Frame-Voyager [3], and the training-free Adaptive Keyframe Sampling (AKS) [4]. We have also added the corresponding performance comparisons, as shown in the updated results table.
>
> Across MVBench, MLVU, and LongVideoBench, our method ZoomV consistently achieves the best performance, demonstrating its effectiveness relative to both training-based and training-free baselines.
>
> |                | MVBench | MLVU | LongVideoBench | VideoMME-long | VideoMME-overall |
> |----------------|---------|------|----------------|---------------|------------------|
> | LongVu         | 66.9    | 65.4 | -              | 59.5          | 60.6             |
> | LongVa         | -       | 56.3 | -              | 45.0          | 52.4             |
> | Frame-Voyager (34B) | 57.3    | 61.1 | -         | 51.2          | 60.0             |
> | AKS            | -       | -    | 62.7           | -             | 65.3             |
> | ZoomV (Qwen-7B ver.)          | 70.3    | 70.0 | 63.3           | 53.9          | 64.4             |
>
> > **Q2**: *Would have appreciate a deeper study on efficiency with a better discussion on training cost/time, inference time versus others methods such as LongVu or other frame selection method such as AKS.*
>
> Inspired by the efficiency analysis in LongVu, we conducted an end-to-end timing study on a 20-minute video understanding task. The results are shown below, and the step-wise breakdown of ZoomV’s runtime can be found in Table 4 of the paper.
>
> | **Models**        | LLaMA-VID | Chat-UniVi | VideoLLaMA2 | VideoChat2 | LLaVA-OneVision | LongVU | ZoomV |
> |-------------------|-----------|------------|-------------|------------|-----------------|--------|-------|
> | Time (sec)        | 55.30      | 49.06      | 58.62       | 45.22      | 25.84           | 32.96  | 12.10   |
>
> It is important to note that LongVu incurs substantial overhead from extracting 1200-frame SigLIP features, which dominates its total inference time. Our method does not require this step, and therefore, avoids this bottleneck. As a result, ZoomV completes the same task in only 12.10s, demonstrating significantly better efficiency compared with LongVu.
>
> > **Q3**: *Would also have liked a more in depth study over the self-reasoning for window selection. Are some models better at that or did you observe the same results for LlaVA-Video, InternVL and Qwen?*
>
> Due to time constraints, we conducted the self-reasoning window selection study using InternVL2.5. The results are shown below:
>
> |                 | VideoMME-long | VideoMME-overall | LongVideoBench | LVBench | MLVU |
> |-----------------|---------------|------------------|----------------|---------|------|
> | InternVL2.5_base  | 52.2          | 63.0             | 60.6         | 41.8    | 67.1 |
> | InternVL2.5_zoomv | 52.6          | 63.1             | 61.6          | 46.6    | 67.3 |
>
> we still observe a +4.8 improvement on LVBench, which is consistent with the conclusions reported in the main paper. This suggests that the effectiveness of our self-reasoning mechanism is not tied to a specific backbone and generalizes well across different video MLLMs.

---

> ### Author Response · Authors · 2025-12-01
> **The response to Reviewer m7v5 (Part 2)**
>
> > **Q5**: *Is there any reason why you did not compare your method with similar training based methods for frame selection?*
>
> We initially selected a subset of representative methods due to space limitations. Following the reviewer’s suggestion, we have now included comparisons with four additional training-based frame selection methods: LongVu [1], LongVa [2], Frame-Voyager [3], VideoChat-R1 [5], and Time-R1 [6]. The corresponding comparative experiments have also been included, as shown in Table 2.
>
> Even under these unfavorable comparison settings, since several of these methods rely on additional grounding-oriented training, ZoomV still outperforms all of them, especially on long-video benchmarks, demonstrating the strong effectiveness of our approach.
>
> ---
>
> [1] Shen, Xiaoqian, et al. "LongVU: Spatiotemporal Adaptive Compression for Long Video-Language Understanding." Forty-second International Conference on Machine Learning.
>
> [2] Zhang, Peiyuan, et al. "Long context transfer from language to vision." arXiv preprint arXiv:2406.16852 (2024).
>
> [3] Yu, Sicheng, et al. "Frame-Voyager: Learning to Query Frames for Video Large Language Models." The Thirteenth International Conference on Learning Representations.
>
> [4] Tang, Xi, et al. "Adaptive keyframe sampling for long video understanding." Proceedings of the Computer Vision and Pattern Recognition Conference. 2025.
>
> [5] Li, Xinhao, et al. "Videochat-r1: Enhancing spatio-temporal perception via reinforcement fine-tuning." arXiv preprint arXiv:2504.06958 (2025).
>
> [6] Wang, Ye, et al. "Time-R1: Post-Training Large Vision Language Model for Temporal Video Grounding." arXiv preprint arXiv:2503.13377 (2025).

---

### Official Review · Reviewer_F8Az · 2025-10-28

**Soundness:** 3
**Presentation:** 3
**Contribution:** 2
**Rating:** 4
**Confidence:** 5

**Summary:**

The paper ZoomV, a query-aware temporal zoom-in framework designed for efficient and accurate long video understanding. It retrieves relevant events and their associated temporal windows as candidates, and select higher-confidence temporal windows as the LVLM's final input to provide the answer. It conducts experiments on temporal grounding benchmarks as well as long video understanding benchmarks to demonstrate the effectiveness of the proposed method.

**Strengths:**

1. The paper is clearly written and easy to follow.
2. The proposed approach is reasonable and methodologically sound.
3. Experiments are conducted on both temporal grounding and long video understanding benchmarks to demonstrate the effectiveness of the method.

**Weaknesses:**

1. One of the main contributions claimed by the paper is the confidence-based temporal grounding approach. However, this concept has already been introduced in TimeSearch [1]. Therefore, it cannot be regarded as a novel contribution of this work. Moreover, the authors have not properly cited TimeSearch to acknowledge prior work.
2. The technical novelty appears limited, as the main modification involves adding textual timestamps to each frame embedding, which was already employed in models such as Eagle2.5 [2] ([Eagle2.5 implementation](https://github.com/NVlabs/Eagle/blob/047e51070e8976978376cb828f7af92323c0f8ef/Eagle2_5/deployment/inference.py#L85))
3. The method seems not consistently effective to all video benchmarks, and the improvement is very trivial in several benchmarks such as MVBench and VideoMME.
4. Since the paper positions its approach as an agent-style method, it should also include comparisons with recent video agent frameworks such as Video-RAG [3].
5. Given that the model is fine-tuned on a recent backbone (Qwen2.5-VL), which already exhibits strong temporal grounding capabilities, it would be more convincing to compare against recent models fine-tuned on the same base, such as VideoChat-R1 [4] and Time-R1 [5].
6. The hierachical search is not a novel idea, which has already explored in TimeSearch [1], UniTime [6] and VideoChat-R1.5 [7]. They should be discussed in the related work and experiments.
7. Since the proposed methods are fundamentally based on temporal grounding, the paper should include a discussion of the temporal grounding task in the related work section.
8. The paper emphasizes efficiency in its title; however, it does not provide a comprehensive analysis of efficiency compared to the base models.

[1] TimeSearch: Hierarchical Video Search with Spotlight and Reflection for Human-like Long Video Understanding, arXiv:2504.01407.

[2] Eagle 2.5: Boosting Long-Context Post-Training for Frontier Vision-Language Models, arXiv:2504.15271.

[3] Video-RAG: Visually-aligned Retrieval-Augmented Long Video Comprehension, NeurIPS 2025.

[4] VideoChat-R1: Enhancing Spatio-Temporal Perception via Reinforcement Fine-Tuning, arXiv:2504.06958.

[5] Time-R1: Post-Training Large Vision Language Model for Temporal Video Grounding, NeurIPS 2025.

[6] Universal Video Temporal Grounding with Generative Multi-modal Large Language Models, arXiv:2506.18883.

[7] VideoChat-R1.5: Visual Test-Time Scaling to Reinforce Multimodal Reasoning by Iterative Perception, arXiv:2509.21100.

**Questions:**

1. The improvement on VideoMME is very limited, for example only 0.1 on Qwen2.5-VL and no improvement on MVBench, while achieves 11.3 on LVBench. It seems that the method is not generalized to all video benchmarks. Could you explain why it achieves improvement by large margin on LVBench, but not effective to VideoMME.
2. Both VideoMME and LVBench are long-video understanding benchmarks that contain thousands of frames. However, the proposed method only samples 64 frames, which results in a substantial loss of visual details throughout the video and may prevent accurate grounding on evidence frames. Have the authors experimented with increasing the number of sampled frames? This could better demonstrate the effectiveness of the proposed approach.
3. The evaluation involves recursively exploring video frames, meaning that the total number of processed frames exceeds 64. How many frames are explored on average?
4. Considering the increased number of processed frames and the computational overhead, is it entirely fair to compare the results with the base model under a 64-frame input setting? A fairer comparison would be against the model’s officially reported best performance, for example, Qwen2.5-VL achieves 70.2 on MLVU and 65.1 on VideoMME.
5. How about the inference efficiency on long video benchmark like VideoMME compared with base models?

---

> ### Author Response · Authors · 2025-12-01
> **The response to Reviewer F8Az (Part 1)**
>
> > **Q1**: *The technical novelty appears limited, as the main modification involves adding textual timestamps to each frame embedding, which was already employed in models such as Eagle2.5.*
>
> Firstly, the use of timestamps in our method is different from that in Eagle 2.5. Specifically, our approach introduces timestamp embeddings at the feature level. In contrast, Eagle 2.5 applies textual timestamps at the prompt level. Moreover, Eagle 2.5 is a concurrent work rather than prior art: its arXiv submission appeared on April 21, while ours was released earlier on April 2.
>
> > **Q2**: *The method seems not consistently effective to all video benchmarks, and the improvement is very trivial in several benchmarks such as MVBench and VideoMME.*
>
> First, our method is primarily designed for long-video understanding, where its advantages are most evident. In addition, when compared with concurrent long-video approaches such as LongVU, we observe a notable contrast on VideoMME: LongVU suffers a significant drop on the short-video subset, while our method maintains stable performance. Although the gains on short-video benchmarks like MVBench and VideoMME are modest, the fact that our method never degrades performance demonstrates its robustness and compatibility across different video lengths.
>
> > **Q3**: *Since the paper positions its approach as an agent-style method, it should also include comparisons with recent video agent frameworks such as Video-RAG.*
>
> First, such a comparison would not be directly fair. Video-RAG leverages additional inputs, including OCR, ASR, and other auxiliary signals, where our method relies only on key-frame visual inputs. The input settings differ substantially, making a direct comparison inappropriate. However, we do include comparisons with other relevant video-agent frameworks, such as VideoChat-R1 and Time-R1, as shown in our response to question 4.
>
> > **Q4**: *Given that the model is fine-tuned on a recent backbone (Qwen2.5-VL), which already exhibits strong temporal grounding capabilities, it would be more convincing to compare against recent models fine-tuned on the same base, such as VideoChat-R1 and Time-R1.*
>
> Thank you for the valuable suggestion. Both VideoChat-R1 and Time-R1 are indeed built upon Qwen2.5-VL, whose temporal grounding capability is already substantially stronger than that of LLaVA-Video. Moreover, VideoChat-R1 and Time-R1 are trained with large amounts of grounding-oriented data, and the training set of VideoChat-R1 even includes the Charades-STA training split.
>
> Despite these advantages, our model based on LLaVA-Video still achieves significantly better performance than both VideoChat-R1 and Time-R1 on ActivityNet-Captions and ReXTime, as shown in the table below.
>
> | Model          | **Charades-STA** |        |        |      | **ActivityNet-Captions** |        |        |      | **ReXTime** |        |      |      |
> | :------------- | :--------------: | :----: | :----: | :--: | :----------------------: | :----: | :----: | :--: | :---------: | :----: | :--: | :--: |
> |                | R@0.3            | R@0.5  | R@0.7  | mIoU | R@0.3                    | R@0.5  | R@0.7  | mIoU | R@0.3       | R@0.5  | mIoU | Acc  |
> | Qwen2.5-VL     | -                | 24.2   | 11.1   | 29.0 | -                        | 15.8   | 7.5    | 21.1 | 17.48       | 11.18  | 15.02| 72.2 |
> | Time-R1        | 78.1             | 60.8   | 35.3   | -    | 58.6                     | 39.0   | 21.4   | -    | 31.81       | 18.46  | 22.48| 72.1 |
> | UniTime        | -                | 59.1   | 31.9   | 52.2 | -                        | 22.8   | 14.1   | 27.3 | -           | -      | -    | -    |
> | VideoChat-R1   | -                | 70.6   | 47.2   | 59.9 | -                        | 33.3   | 16.7   | 35.5 | 42.56       | 28.45  | 29.99| 71.8 |
> | ZoomV          | 73.6             | 52.4   | 24.5   | 48.6 | 61.0                     | 43.0   | 26.1   | 43.9 | 48.4        | 36.4   | 36.7 | 76.5 |
>
>
> > **Q5**: *The hierachical search is not a novel idea, which has already explored in UniTime and VideoChat-R1.5. They should be discussed in the related work and experiments.*
>
> Thank you for the thoughtful suggestion. We have added a detailed discussion of UniTime [6] and VideoChat-R1.5 [7] in the revised related work section, explaining the key differences between their hierarchical search strategies and ours. The corresponding comparative experiments have also been included, as shown in Table 2.
>
> > **Q6**:  *Since the proposed methods are fundamentally based on temporal grounding, the paper should include a discussion of the temporal grounding task in the related work section.*
>
> Thank you for the valuable suggestion. We have incorporated a dedicated discussion of temporal grounding into the related work section in the revised version of the paper.

---

> ### Author Response · Authors · 2025-12-01
> **The response to Reviewer F8Az (Part 2)**
>
> > **Q7**: *The paper emphasizes efficiency in its title; however, it does not provide a comprehensive analysis of efficiency compared to the base models.*
>
> First, we would like to clarify that the efficiency highlighted in our paper refers specifically to comparisons against video-agent–style methods. In Section 4.3, we already provide a detailed latency analysis against Video-Tree, showing that our method achieves a 30.8% lower inference delay while delivering higher accuracy at the same time.
>
> > **Q8**: *The improvement on VideoMME is very limited, for example only 0.1 on Qwen2.5-VL and no improvement on MVBench, while achieves 11.3 on LVBench. It seems that the method is not generalized to all video benchmarks. Could you explain why it achieves improvement by large margin on LVBench, but not effective to VideoMME.*
>
> For VideoMME, we observe that the improvements mainly appear in the perception-oriented tasks, particularly in OCR-related evaluations, where our temporal-light mechanism helps the model focus more effectively on key frames. In contrast, the reasoning tasks in VideoMME generally show slight performance drops.
>
> | Task                   | Internvl3 | ZoomV  |
> |------------------------|-----------|--------|
> | Perception & Recognition | 69.7      | **70.6** |
> | OCR Problems           | 71.9      | **74.1** |
> | Reasoning              | 64.1      | 63.3   |
> | Information Synopsis   | 79.0%     | **79.6** |
>
> For LVBench, the situation is different. First, our zoom-in mechanism is highly effective for long-video scenarios, enabling more accurate retrieval of informative segments. Second, LVBench predominantly consists of perception and localization tasks, for which our method is especially well suited. As a result, the gains on LVBench are significantly larger.
>
> | Task                    | Internvl3 | ZoomV  |
> |--------|-----------|--------|
> | key information retrieval | 44.5      | **58.1** |
> | event understanding      | 41.5      | **47.6** |
> | entity recognition       | 44.3      | **54.1** |
> | reasoning                | 46.7      | **52.2** |
> | temporal grounding       | 35.9      | **43.6** |
> | summarization            | 37.3      | 39.6   |
>
> > **Q9**: *Both VideoMME and LVBench are long-video understanding benchmarks that contain thousands of frames. However, the proposed method only samples 64 frames, which results in a substantial loss of visual details throughout the video and may prevent accurate grounding on evidence frames. Have the authors experimented with increasing the number of sampled frames? This could better demonstrate the effectiveness of the proposed approach.*
>
> As shown in the table below, increasing the number of sampled frames does not necessarily yield higher accuracy. This is because ZoomV inherently possesses strong temporal grounding capabilities; adding more frames can introduce additional noise and even interfere with the grounding process. These results further demonstrate that our method can maintain competitive performance even with only 64 sampled frames.
>
> |       | VideoMME-long | LongVideoBench | LVBench | MLVU |
> |-------|---------------|----------------|---------|------|
> | 64  frames  | 53.9          | 63.3           | 51.5    | 70.0 |
> | 72  frames  | 54.4          | 62.3           | 51.1    | 69.8 |
> | 80   frames | 54.4          | 62.2           | 50.9    | 69.4 |
>
>
> > **Q10**: *The evaluation involves recursively exploring video frames, meaning that the total number of processed frames exceeds 64. How many frames are explored on average?*
>
> Yes, our method performs multiple rounds of corrective searching, but each round explores only 16 additional frames. Therefore, the total number of explored frames is computed as:
>
> Frames Explored = 64 (preview frames) + search_rounds * 16
>
> The average explored frame count for each benchmark is reported in the table below.
>
> |      | VideoMME-long | VideoMME-overall | LVBench | MLVU | LongVideoBench |
> |------|---------------|------------------|---------|------|----------------|
> | Avg. Frames | 90            | 89               | 104     | 91   | 89             |
>
> Importantly, only the 64 selected frames are passed to the LLM for final inference; the additional explored frames are used solely for search and do not increase the LLM-side computational cost.

---

> ### Author Response · Authors · 2025-12-01
> **The response to Reviewer F8Az (Part 3)**
>
> > **Q11**: *Considering the increased number of processed frames and the computational overhead, is it entirely fair to compare the results with the base model under a 64-frame input setting? A fairer comparison would be against the model’s officially reported best performance, for example, Qwen2.5-VL achieves 70.2 on MLVU and 65.1 on VideoMME.*
>
> We believe this concern arises from the assumption that processing more frames substantially increases computational cost. In practice, however, the dominant computation lies in the LLM stage, not in the visual encoder. The additional cost introduced by our extra frame processing is less than 1\% relative to the full inference pipeline. Moreover, our method completes token selection before entering the LLM, so the input to the language model remains nearly identical to that of the 64-frame baseline. Therefore, the comparison is conducted under effectively aligned computational settings.
>
> > **Q12**: *How about the inference efficiency on long video benchmark like VideoMME compared with base models?*
>
> Thank you for the suggestion. We have compared the inference efficiency not only on VideoMME, but also on MLVU and LongVideoBench, using the respective base models as references. Although our method introduces a slight increase in latency due to the incorporation of multi-turn calibration dialogs, this design substantially improves video understanding accuracy—for example, yielding a +3.7 improvement on MLVU.
>
> |           | VideoMME (s) |MLVU (s) | LongVideoBench (s)|
> |--------|-----------------|---------|------|
> | BaseModel (LLaVA-Video)   | 1.8          | 1.9  | 1.9      |
> | w/ ZoomV        | 3.1            | 3.4 | 3.1  |
>
> More importantly, when compared with video-agent–style methods, our approach is significantly more efficient. As shown in Section 4.3, our method achieves a 30.8% lower inference delay than Video-Tree while simultaneously providing higher accuracy.

---

### Author Response · Authors · 2025-12-02
**Summary of Common Concerns on Paper 3376**

Dear Area Chairs,

Thank you for coordinating the review process of paper 3376. We sincerely appreciate the constructive feedback of reviewers, which helps us improve the work. Below, we summarize the common concerns raised by the reviewers and provide consolidated responses.

**1. The efficiency of ZoomV**

> Proposed by Reviewer **F8Az**, **m7v5**, and **K5CS**.

First, we would like to clarify that the **efficiency** highlighted in our paper refers specifically to comparisons against **video-agent–style methods**. In Section 4.3, we already provide a detailed latency analysis against **Video-Tree**, showing that our method achieves a **30.8% lower inference delay** while delivering **higher accuracy** at the same time.

Besides, following the efficiency analysis in **LongVU** [1], we conduct an end-to-end timing study on a **20-minute video understanding task**. It is important to note that LongVU incurs substantial overhead from extracting 1200-frame SigLIP features, which dominates its inference time. Our method avoids this bottleneck and completes the same task in **12.10s**, demonstrating significantly **better efficiency** compared with LongVU.

| **Models**        | LLaMA-VID | Chat-UniVi | VideoLLaMA2 | VideoChat2 | LLaVA-OneVision | LongVU | ZoomV |
|-------------------|-----------|------------|-------------|------------|-----------------|--------|-------|
| Time (sec)        | 55.3      | 49.06      | 58.62       | 45.22      | 25.84           | 32.96  | 12.10  |

Finally, we comprehensively compare the inference time of ZoomV with other agents and long-video models **across varying durations**, as illustrated in Figure 7 in the revision. These results underscore the superior scalability and efficiency of  ZoomV for long-video understanding.

**2. The comparison of training-based methods**

> Proposed by Reviewer **F8Az**, **m7v5**, and **K5CS**.

Following the reviewer’s suggestion, we have now included comparisons with **five additional training-based frame selection methods**: LongVU [1], LongVa [2], Frame-Voyager [3], VideoChat-R1 [4], and Time-R1 [5] in Tables 1 and 2. **Even under these unfavorable comparison settings**, since several of these methods rely on **additional grounding-oriented training**, ZoomV (Qwen ver.) still **outperforms all of them**, especially on long-video benchmarks, demonstrating the strong effectiveness of our approach.

**3. The performance on short-video benchmarks**

> Proposed by Reviewer **F8Az**.

First, our method is primarily **designed for long-video understanding**, where its advantages are most evident. In addition, when compared with concurrent long-video approaches such as LongVU, we observe a notable contrast on VideoMME: LongVU suffers **a significant drop** on the short-video subset, while our method **maintains stable performance**. Although the gains on short-video benchmarks like MVBench and VideoMME are modest, the fact that our method never degrades performance demonstrates its robustness and compatibility across different video lengths.

---

[1] *Shen, Xiaoqian, et al. "LongVU: Spatiotemporal Adaptive Compression for Long Video-Language Understanding." Forty-second International Conference on Machine Learning.*

[2] *Zhang, Peiyuan, et al. "Long context transfer from language to vision." arXiv preprint arXiv:2406.16852 (2024).*

[3] *Yu, Sicheng, et al. "Frame-Voyager: Learning to Query Frames for Video Large Language Models." The Thirteenth International Conference on Learning Representations.*

[4] *Li, Xinhao, et al. "Videochat-r1: Enhancing spatio-temporal perception via reinforcement fine-tuning." arXiv preprint arXiv:2504.06958 (2025).*

[5] *Wang, Ye, et al. "Time-R1: Post-Training Large Vision Language Model for Temporal Video Grounding." arXiv preprint arXiv:2503.13377 (2025).*

---

### Author Response · Authors · 2025-12-03
**General Response and Revision of Manuscript**

Dear Area Chairs,

Thank you for coordinating the review process of paper 3376.
We sincerely thank the reviewers for their valuable feedback!

In this post:

1. We summarize and highlight the key strengths of our work as noted by the reviewers.
2. We summarize the changes to the updated PDF document (highlighted in blue).


**1. Key Strengths**

**Motivation and Method**

> `K5CS`: "End-to-end single-model design which is a very important point for video agents."; "a novel human-inspired framework"
>
> `m7v5`: "self-correction to select the frames is an interesting idea"; "TemporaLink and TemporaLight are sounded and relevant";
>
> `F8Az`: "reasonable and methodologically sound."


**Experiments and Results**


> `F8Az`: "on both temporal grounding and long video understanding benchmarks"
>
> `m7v5`: "use method on top of LlaVa, InternVL and Qwen2.5-VL"; "improvement on MVBench, MLVU, LongVideoBench, VideoMME, LVBench, Charades-STA, ActivityNet-Caption and ReXTime"
>
> `K5CS`: "state-of-the-art results"; "short-video performance is *not sacrificed*"; "ablation studies on how their modules can be beneficial for long video understanding"

**Presentation**

> `K5CS`: "motivation is clear and well defined"
>
> `F8Az`: "clearly written and easy to follow."
>
> `m7v5`: "well written"

**2. Changes to the PDF**

**Related Work**

- `F8Az`: Add a dedicated discussion of temporal grounding into the related work section.

- `m7v5`: Add detailed discussions of the mentioned long-video models and frame selection methods, including LongVu, Frame-Voyager, and Adaptive Keyframe Sampling (AKS).

- `F8Az`: Add a detailed discussion of UniTime and VideoChat-R1.5 in the revised related work section, explaining the key differences between their hierarchical search strategies and ours.


**Experiments**

- `F8Az`, `m7v5`, `K5CS`: Include comparisons with additional methods: Qwen2.5VL, LongVU, LongVa, VideoChat-R1, and Time-R1 in Tables 1 and 2.

- `K5CS`: Evaluate ZoomV on InternVL3 in Table 1.

**Efficiency**

- `F8Az`, `m7v5`, and `K5CS`: Comprehensively compare the inference time of ZoomV with other agents and long-video models across varying durations in **Figure 7 in Section 4.3**.


**Method**

- `K5CS`: We have revised Section 3.3 by adding clearer explanations of the key steps, intermediate reasoning, and overall workflow.

---

### Meta-Review · Area_Chair_PBpk · 2026-01-07

**Summary:**

## Summary
This paper introduces ZoomV, an agentic-based approach for long video understanding. Experiments are done on both long and short-video understanding benchmarks where larger improvements are observed on long-video benchmark. Experiments are also done in video temporal grounding task.

Overall, the novelty is moderate (because of Eagle2.5, UniTime, VideoChat-R1.5), experiments are OK but not super strong. ZoomV provides a larger improvements on LongVideoBench but slim improvements on other benchmarks.

## Discussion & Rebuttal
* [In response to reviewer F8Az] On inconsistent performance across different datasets, the author(s) provided the breakdown (per category) performance for VideoMME, the answer cannot clear the concern about this inconsistency.  The answer also does not provide any insight why it happens.

* [In response to reviewer m7v5] On missing baselines, the rebuttal provides additional comparisons with LongVu, LongVa, Frame-Voyager, and AKS. ZoomV has the best proformance on MVBench, MLVU, and 	LongVideoBench, and yet still has some worse wrt comparing methods: e.g. VideoMME-long -5.6%, VideoMME-overall: -0.9%.

* In response to the concern about the efficiency analysis, the authors provides some additional inference time comparisons. Overall, the proposed method is faster than other video-agent-based methods, but still slower than its base model.

## Decision
AC reads all reviews and rebuttal QAs. AC appreciates the additional efforts the author(s) had put into the rebuttal. Although AC believes the paper has the potential, AC finds the paper has a few concerns that had not been fully addressed or understood. AC recommends to reject this paper at its current form.

**Reviewer Concerns:**

## Concern from Reviewers
*  Limited technical novelty w.r.t other methods (e.g., Eagle2.5, UniTime, VideoChat-R1.5).
* The proposed method is not consistent across benchmarks, and the improvement is slim in several benchmarks such as MVBench and VideoMME.
* A number of missing baselines (LongVu, LongVa, Frame-Voyager and AKS).
* Missing a deeper analysis on efficiency on training cost/time, inference time.

**Reviewer Scores:**

Reviewer scores are initially given as 4, 4, and 6. No reviewer indicates that they are happy with the provided answers during rebuttal.

---

### Decision · Program_Chairs · 2026-01-26

Reject